# Longitudinal Associations between Sensation Seeking and Its Components and Alcohol Use in Young SWISS Men—Are There Bidirectional Associations?

**DOI:** 10.3390/ijerph191912475

**Published:** 2022-09-30

**Authors:** Gerhard Gmel, Simon Marmet, Nicolas Bertholet, Matthias Wicki, Joseph Studer

**Affiliations:** 1Addiction Medicine, Lausanne University Hospital and University of Lausanne, Rue du Bugnon 23A, 1011 Lausanne, Switzerland; 2Research Department, Addiction Switzerland, Avenue Louis-Ruchonnet 14, 1003 Lausanne, Switzerland; 3Centre for Addiction and Mental Health, Institute for Mental Health Policy Research, 250 College Street, Toronto, ON M5T 1R8, Canada; 4Alcohol and Research Unit, University of the West of England, Frenchay Campus, Coldharbour Lane, Bristol BS16 1QY, UK; 5School of Social Work, University of Applied Sciences and Arts Northwestern Switzerland, 4600 Olten, Switzerland; 6Institute for Research, Development and Evaluation, Bern University of Teacher Education, 3012 Bern, Switzerland; 7Service of Adult Psychiatry North-West, Department of Psychiatry, Lausanne University Hospital, Chemin des Chaux, 1196 Prangins, Switzerland

**Keywords:** personality traits, sensation seeking, alcohol use, latent change scores, cross-lagged effects, young men

## Abstract

The association between alcohol use and sensation seeking is well known. Less is known about whether longitudinal changes in alcohol use are associated with changes in sensation seeking and in which direction influence might flow. 5125 men aged 20.0 years old at baseline and 25.4 years old at follow-up responded to the Brief Sensation Seeking Questionnaire, which measures four subscales of experience seeking, boredom susceptibility, thrill- and adventure-seeking, and disinhibition. Alcohol use was measured using volume (drinks per week) and binge drinking (about 60 g or more per occasion). Associations were calculated using cross-lagged panel models and two-wave latent change score models. Correlations between the latent change scores for alcohol use and the sensation-seeking subscales were all positive, being largest for disinhibition (r > 0.3) and much smaller (r ~ 0.1) for the others. Disinhibition was the dominant effect over the entire sensation-seeking scale. Cross-lagged paths were (except for thrill- and adventure-seeking) bidirectional and mostly higher from alcohol use to sensation seeking (e.g., path_volume-disinhibition_ = 0.136, and path_disinhibition-volume_ = 0.072). Again, effects were highest for disinhibition. Given the bidirectional links between sensation seeking and alcohol use, preventive efforts aiming to achieve stable positive changes in alcohol use and personality should target both simultaneously and focus on disinhibition.

## 1. Introduction

Several reviews have shown that alcohol use is cross-sectionally associated with personality traits such as neuroticism, extraversion, impulsivity, and sensation seeking [1,2,3,4,5]. Personality traits longitudinally predicted alcohol use. Higher neuroticism, higher extraversion, lower conscientiousness, and lower agreeableness were associated with higher alcohol use over a nine-year period [6]. Studies on the inverse association, i.e., whether alcohol use and changes in alcohol use predict changes in personality, are rare. Bidirectional associations [7] could create a vicious circle between alcohol use and a particular personality trait. Although both internalizing behaviors, such as depression, and externalizing behaviors, such as aggression have been associated with alcohol use [8,9], externalizing personality traits, and particularly the broad trait of impulsivity, may be the most relevant to alcohol use [3,10,11]. In an earlier study on the same cohort, Gmel and colleagues [5] looked at such bidirectional longitudinal associations between changes in four personality traits (aggression–hostility, sociability, neuroticism–anxiety, and sensation seeking as impulsivity trait) and changes in alcohol use (volume of drinking and binge drinking). They found that the full scale of sensation seeking had the strongest cross-sectional correlation with both alcohol measures, the strongest correlation between change in personality and change in both alcohol use measures, and the strongest bidirectional longitudinal associations with both alcohol measures. A meta-analytical review [3] considering the subscales of impulsivity (i.e., urgency, lack of premeditation, lack of perseverance, and sensation seeking (UPPS): e.g., Ref. [12] showed that sensation seeking was the dimension most strongly associated with alcohol consumption in terms of quantity, frequency, and binge drinking (heavy drinking occasions). Gmel and colleagues [5] used the conceptualization of sensation seeking as suggested by Zuckerman et al. [13] or Hoyle et al. [14] in its short form as a total scale. The total scale comprises the four subconstructs of experience seeking, boredom susceptibility, thrill and adventure seeking, and disinhibition. However, the conceptualization of sensation seeking in other models like the UPPS model is different. In attempts to better classify different impulsigenic personality traits as in the UPPS model, disinhibition and boredom susceptibility were grouped together but under the category of (lack of) perseverance whereas only facets such as experience seeking and thrill and adventure seeking were seen as sensation seeking [3,12,15]. As outlined by Whiteside and Lynham [12], the “jingle” and “jangle” fallacies, i.e., that two construct with relatively equivalent labels may be rather different, whereas two other constructs with different labels may be rather equivalent, is very present in the research on impulsivity in general and sensation seeking in particular. Thus, findings with one conceptualization of sensation seeking my lead to other results as with another conceptualization, and it is important to disentangle the facets of subscales and not only to look at total scales as done by Gmel and colleagues [5] to understand apparent discrepancies between studies using different conceptualizations of what is labelled “sensation seeking”. The study by Gmel and colleagues [5] was designed to compare different personality traits and not to investigate facets of impulsivity only. The present study examined the four subconstructs of sensation seeking as defined by Zuckerman, Eysenck and Eysenck [13] longitudinally over five-years during emerging adulthood (baseline measurement at 20 years old). Particularly, the study looks at the bidirectional associations between alcohol use and sensation seeking subconstructs and the associations between changes in alcohol use and changes in sensation-seeking subconstructs.

Sensation seeking has long been measured and conceptualized with the Sensation Seeking Scale (SSS) Form V [13]. Because of its length, a shorter form, the brief (B)SSS was developed by Hoyle et al. [14]. It is a multifaceted yet relatively stable personality construct consisting of experience seeking, boredom susceptibility, thrill- and adventure-seeking, and disinhibition [14,16,17]. Experience seeking describes the interest in new experiences for its own sake. These can be obtained by travelling to uncommon places, listening to modern, arousing music, meeting with unconventional persons, but also the use of marihuana or hallucinogenic drugs. People highly susceptible to boredom frequently report an intolerance of repetition and routine and feeling restless. They feel bored even in situation that others may find stimulating. Thrill- and adventure-seeking is characterized by a desire to engage in physical, commonly outdoor activities that involve speed or danger. Finally, disinhibition is a preference for activities, which can get out of control, such as wild parties involving reduced social restraint. Disinhibited individuals have less control over their impulses. This may lead to experimental behaviors that are inappropriate for the situations that people are in. Disinhibition may include having many sexual partners, may increases gambling, illicit drug use, alcohol use, and engagment in illegal activities. A meta-analysis by Hittner and Swickert [17] indicated that of the four sensation-seeking subconstructs, disinhibition had the strongest association with alcohol use. A recent longitudinal study among college students found reciprocally reinforcing effects between disinhibition and drinking quantity, anticipatory effects of thrill- and adventure-seeking for drinking frequency and quantity, but no temporal directionality for boredom susceptibility and experience-seeking [18]. These findings showed that subscales of sensation seeking might have differential associations with alcohol use. Moreover, the different conceptualizations of sensation seeking may lead to “jingle” and “jangle” fallacies. For example, a longitudinal study of Riley et al. [19] using the UPPS conceptualization could not find any meaningful change in sensation seeking among college freshman, and therefore the authors did not model its association with alcohol use. Sensation seeking was defined by thrill and adventure seeking. However, the authors found that alcohol problems predicted changes in urgency, lack of planning, and lack of perseverance, which contains aspects of sensation seeking (e.g., disinhibition or boredom susceptibility) as defined by Zuckermann et al. [13]. Therefore, besides the likelihood of differential effects on different aspects of alcohol use, there is a need to shed a light on subconstructs and not only the total scale to identify those factors particularly relevant for the link with alcohol use. 

Personality trait differences between individuals are seen as relatively stable (rank-order consistency) over time [20]. Nevertheless, normative mean-level changes in personality traits can occur across a life course. Some may grow between adolescence and middle adulthood, e.g., agreeableness and conscientiousness [21,22]. Extraversion may decline [23], and sensation seeking normatively declines from about 15 years old [3]. Research on sensation seeking and alcohol use is often limited to adolescence, perhaps because that developmental period is characterized by impulsive decision making [3], which is particularly relevant for sensation seeking and related concepts like experience seeking or fun seeking [16,24,25]. There seems to be a link between the development of subcortical reward-processing areas and of the prefrontal cortex and sensation seeking and impulsivity [26]. These aspects of brain development have already significantly advanced in adolescence, and therefore alcohol use in adulthood may be less important for predicting them. Similarly, alcohol use becomes more normative with age and therefore may be more relevant to early adolescent drinking and predicting later alcohol use than looking at personality development and alcohol use in adulthood. Regarding impulsigenic personality traits, longitudinal risk research has focused on predicting alcohol use and rarely on predicting impulsigenic traits from alcohol use [27]. Nevertheless, neural maturation continues into the early twenties [26]. As Stautz and Cooper [3] argued, individuals legally considered adults because of their age may still be adolescents in terms of their neurodevelopment, and thus the inverse direction from alcohol use to sensation seeking may be similarly important.

Nevertheless, personality traits may also go through repeated short-term change sequences. The TESSERA framework, among others, suggests this (Triggering situations, Expectancy, States/State expressions, and Reactions: [22]). TESSERA contains many aspects, including triggering situations, expectancy, and associative processes (e.g., implicit learning, reinforcement learning, habit formation). Alcohol use may reflect such repeated short-term change sequences through reinforcement and reward. This could lead to changes in personality traits, particularly sensation seeking. Such personality changes are also plausible along biomedical pathways (see [17]). Sensation seeking is negatively correlated with platelet levels of monoamine oxidase (MAO) [16]. MAO has also an influence on dopamine levels stimulating dopamine release [28,29]. Dopamine levels are associated with sensation seeking as they are connected with reward-seeking behavior [16]. Thus, the link between impulsivity in general (especially sensation seeking) and alcohol use should form a bidirectional process: high levels of (subconstructs of) sensation seeking may lead to higher levels of alcohol use, and this could be reinforced due to the neurobiological changes caused by alcohol [3].

Given this likely bidirectional association, surprisingly few studies have examined whether changes in alcohol use are related to changes in sensation seeking (or personality in general), or whether alcohol use predicts subsequent changes in sensation seeking. Besides their theoretical and biomedical relevance, correlated changes might also have clinical relevance. As Littlefield, Sher, and Woods [30,31] argued, concurrent post-treatment changes may indicate a substantial and persistent underlying change. A personality change may then support a long-term change in alcohol use. However, it may be predictive of a future relapse and a recurrence of heavy alcohol use if individuals changed their alcohol use but their personality did not change. A correlated change without directional change may indicate a third-variable explanation (e.g., normative changes or genetic predisposition). The direction of effects (from or to personality) may also inform treatment approaches. If the longitudinal association is predominantly from alcohol to personality, then the focus should be on reducing alcohol use to avoid any deterioration towards risky personality profiles. If the predominant direction is the other way around, then treatments targeting personality may be a promising approach. Individually tailoring the prevention of alcohol use problems to personality traits has shown long-term effectiveness in different countries [32,33,34].

A few studies have looked at the impact of alcohol use on personality across the broad range of extraversion types, the narrower range of impulsivity, and, finally, specifically for sensation seeking. As stated above, most studies were on (older) adolescents, often college student samples. College students experience a significantly lower degree of parental monitoring and more freedom over their behavior than do non-college students [3]. Additionally, US college students often have a highly advantageous socio-economic status such that college samples may be biased towards more affluent groups of older adolescents. One study looked at transitions from high school, through college, and post-college [35]. Decreases in binge drinking generally paralleled commonly normative decreases in impulsivity and sensation seeking. However, the group that showed high and increasing binge drinking levels during their college years showed non-normative increases in sensation seeking and impulsivity during their transition out of college. Hicks, Durbin, Blonigen, Iacono, and McGue [36] looked at alcohol use disorder (AUD) and behavioral disinhibition from late adolescence (17 years old) to emerging adulthood (24 years old). Individuals who overcame their baseline AUD at follow-up exhibited less behavioral disinhibition at 24 years old than did those with a persisting AUD. Indeed, they almost returned to the levels of individuals who had never had an AUD. The authors concluded that the course of AUDs might affect changes in personality. Kaiser, Davis, Milich, Smith, and Charnigo [7] looked at college students aged around 19 years old at baseline and followed-up for three years. They found that sensation seeking and alcohol use bidirectionally reinforced each other. This confirmed findings by White et al. [37] in adolescence and by Quinn, Stappenbeck, and Fromme [38] for the transition from high school and on through college years. Regarding the direction of effects, in their college sample, Littlefield et al. [31] found that correlated changes between alcohol use and novelty-seeking were better explained by the cross-lagged effect of personality influencing later alcohol use, rather than vice versa.

A few studies have looked at personality changes in older populations. Hakulinen and Jokela [39] used different samples, with a mean age of 51.5 years old (and a mean follow-up of 5.6 years), and Luchetti, Terracciano, and Sutin [40] used latent difference score models to follow people in middle and older adulthood (50 years old and above) over four years. Even in these populations, baseline heavy alcohol use was associated with increasing extraversion, and changes from risky to non-risky alcohol use during follow-up were associated with decreasing extraversion. 

The present study looked at bidirectional associations between alcohol use and sensation seeking as the total scale and the four subconstructs of experience seeking, boredom susceptibility, thrill- and adventure-seeking, and disinhibition, in a five-year follow-up study of young men aged 20 years old at baseline and from a non-selective sample regarding socio-economic status or affluence. Changes in personality were compared with changes in drinking volume and binge drinking. In addition, cross-lagged associations from personality to alcohol use and from alcohol use to personality were estimated. We hypothesized that correlations would be positive and strongest for disinhibition [17]. There has been little research regarding the direction of effects. Theoretically, bidirectional links would be expected, but the link from sensation seeking to alcohol use would be expected to be stronger than vice versa.

## 2. Methods

### 2.1. Participants

Data came from the longitudinal Cohort Study on Substance Use Risk Factors (C-SURF). Not all questions were asked at every study wave to avoid an overly heavy response burden. Personality was deemed relatively stable and was therefore only examined in the first (hereafter baseline) and third (hereafter follow-up) waves. To determine their eligibility for service in military or civil defense units, virtually all Swiss men must attend a mandatory recruitment procedure at around 19 years old. These two-day procedures were used to enroll conscripts for C-SURF. It is important to note that participation in C-SURF was completely independent of these procedures and service in military or civil defense units. Assessments were completed at home, via the internet, and the military was not informed about responses. Those who preferred could fill out a paper version of the questionnaire. C-SURF was approved by the Human Research Ethics Committee of the Canton of Vaud (Protocol No. 15/07). 

During enrollment procedures, 15,066 Swiss men showed up at the Army recruitment centers and were eligible for study inclusion. However, officers of the army were in charge to inform conscripts about the study. This was forgotten for few recruitment days resulting in a loss of 1829 potential participants, who could not be approached for study participation. The loss is likely to be random. Baseline assessment occurred between September 2010 and March 2012: 7556 conscripts provided their written informed consent to participate, and 5987 (79.2%) completed the questionnaire. Follow-up took place between April 2016 and March 2018: of those who completed the baseline assessment, 5125 (85.6%) completed the follow-up. Mplus software (version 8.1) was used for the main analyses (latent change scores, see below). Maximum likelihood estimations with robust standard errors were used to account for skewness in the observed variables. Full information maximum likelihood (FIML) estimators were used, which enabled the inclusion of the few participants with missing values using the missing at random assumption. For descriptive purposes, the tables included use the available data with their various missing values and indicating corresponding sample sizes. 

### 2.2. Measurements

#### 2.2.1. Sensation Seeking

Sensation seeking was measured using the Brief Sensation Seeking Scale (BSSS: [14]), with eight items on a five-point Likert scale (from “strongly agree” to “strongly disagree”) for the full sensation-seeking score, i.e., two items each for the four subscales, namely for experience-seeking, boredom susceptibility, thrill and adventure seeking, and disinhibition. Full scale mean scores were used if at least six items were answered. Means were then scaled up to the original metric (sums). For subscale scores to be valid, both items had to be answered. One major advantage of the BSSS, besides its brevity, is that its items do not directly ask about alcohol or drug use or the desire to use them, as do many other instruments on sensation seeking or impulsivity. Indeed, this would create predictor-criterion contamination [41] when analyzed with, e.g., alcohol use as an outcome. Such contaminated scales are against standard epidemiological norms that measure exposure and outcomes independently [42]. The social science literature has debated predictor-criterion contamination, particularly with regard to alcohol use and sensation seeking [43,44,45], because estimations of subscales such as disinhibition may be exaggerated or partly tautological if they contain the notion of alcohol use directly [46] when establishing a priori causal directions.

#### 2.2.2. Alcohol Use Measurements

Respondents’ usual number of drinking days over the past 12 months were examined separately for weekends (Friday, Saturday, and Sunday) and weekdays (Monday–Thursday). Potential quantities per drinking day were 1–2 drinks, 3–4 drinks, 5–6 drinks, 7–8 drinks, 9–11 drinks, and 12 drinks or more. Frequencies and quantities were combined to yield a number of drinks per week as a volume-of-drinking measurement. This variable was log-transformed due to its skewness. One drink was added before taking the logarithm because the log of 0 is not defined and adding one drink puts the minimum value of non-drinkers to zero with the logarithmic transform [47].

We defined binge drinking, using the standard question from the Alcohol Use Disorders Identification Test (AUDIT), as the consumption of six drinks or more (approximately 60 g or more of pure alcohol) on one occasion in the past 12 months (response options from 0 to 4): never, less than monthly, monthly, weekly, and daily or almost daily). We used the original rating scale because converting into annual days of binge drinking had a big impact on estimations due to rare daily binge drinkers acting as statistical outliers. No other questions of the AUDIT were asked. Measurements for volume and binge drinking provided higher estimates than the original unlogged weekly number of drinks or the binge drinking variable when converted into annual binge drinking days. However, rank order of correlations with sensation seeking across the sensation-seeking (sub-) scales did not change. Thus, the general interpretation of findings would not be changed by coding alcohol use variables differently. 

### 2.3. Statistical Analysis

Changes between baseline and follow-up were tested using paired *t*-tests. Pearson’s correlations were used for descriptive sample correlations with observed variables. The main statistical model used was the cross-domain coupling model [48], a latent change score model. Figure 1 shows the basic model, which was estimated separately for the total BSSS score and its subscales, and for both binge drinking and drinking volume. βs can be interpreted as cross-lagged coefficients (coupling parameters). They indicate, whether subscales of sensation seeking and the full scale (baseline) predict alcohol use change scores (follow-up) or whether alcohol use (baseline) predicts personality change scores. The difference between the cross-lagged was tested by fixing the coefficients to be equal and comparing the model fit versus the unrestricted model. ∆s represent the latent change scores. γs are the paths from baseline scores to corresponding change scores, and they adjust for regression to the mean, which commonly occurs because individuals with high baseline scores tend to have lower follow-up scores on the same construct, and vice versa. This results in a negative association between the initial status and the change score [49]. Finally, *ρ* is the correlation between the latent change scores, indicating how strongly changes in sensation seeking and alcohol use correlate, while adjusting for regression to the mean and considering coupling pathways (cross-lagged paths). As there were ten models (four subscales of sensation seeking and the total scale with two measures of alcohol use), significances may be influenced by multiple comparisons. The need for adjustment of multiple testing is debated in the literature [50,51,52]. We therefore reported unadjusted *p*-values. However, adjusted *p*-values for the conservative Bonferroni adjustment can be obtained by taking tenfold the presented *p*-values. In general, almost all significant findings in the cross-domain coupling model would remain significant (*p* < 0.05) after Bonferroni adjustment. We refer to effects if they would be close to or exceeding the Bonferroni adjusted *p* of 0.05.

Latent change score models have the advantage of reducing measurement errors better than models that analyze changes in observed variables. We nevertheless calculated two additional sets of correlations as a sensitivity analysis: the correlation between changes in sensation seeking and alcohol use in observed variables (follow-up measurements minus baseline measurements). We also used the residuals from the two regressions (one for alcohol and one for each personality trait) of the changes between observed baseline and follow-up measurements on the baseline measurements. The correlation of these residuals account for regression to the means in observed variables. Finally, simple cross-lagged panel models [53] of the observed variables (see Figure 2) were calculated as a sensitivity analysis. 

Correlation coefficients of 0.1 are considered small effect sizes, those of 0.3 are considered medium, and those of 0.5 are considered large [54]. For standardized β values, we used the conversion to correlations proposed by Peterson and Brown [55] to obtain approximate effect sizes, i.e., r = 0.98 β + 0.05 λ, where λ is an indicator variable equal to 1 when β is non-negative and equal to 0 when β is negative.

## 3. Results

Participants were aged 20.0 (standard deviation (SD) = 1.25) years old on average at baseline and 25.4 (SD = 1.22) years old at follow-up. The strongest cross-sectional correlations with drinking volume and binge drinking among the four sensation-seeking subscales were found for disinhibition. Correlations with the three other subscales were smaller but positive (Table 1). The total BSSS score correlation for sensation seeking lay between disinhibition and the remaining subscales, and all the correlations were significant (*p* < 0.001). 

Drinking volume and binge drinking declined (Table 2). On the non-logged scale, this corresponded to a decrease in volume of about one drink per week. The change on the rating scale, presented in Table 2, would mean an annual reduction of about nine binge drinking occasions. Disinhibition showed the expected normative decline from baseline to follow-up and accounted for most of the change in the total sensation-seeking scale score. Similarly, thrill-seeking declined, whereas experience seeking and boredom susceptibility and experience seeking increased. The initial values and change values were negatively correlated. This was true for both alcohol use measurements and sensation-seeking subscales, pointing to effects of regression to the mean (Table 2). 

Table 3 presents the latent change score model correlations between changes in alcohol use measurements and sensation-seeking measurements (right panel; *ρ* in Figure 1). For comparison, we also calculated the correlations between observed changes, unadjusted (left panel) and adjusted (middle panel) for regression to the mean. Models considering regression to the mean (latent and observed changes) commonly calculated higher correlations, with disinhibition showing the highest correlations, with medium effect sizes (r > 0.3) for both the latent change score model and regression to the mean adjusted correlations, and for both drinking volume and binge drinking. All correlations were positive and significant (*p* < 0.001), including for the three other subscales of experience-seeking, boredom susceptibility and thrill-seeking, but much lower, not even reaching a small effect size (r < 0.1) for thrill- and adventure-seeking. 

The latent change score model’s standardized cross-lagged (coupling) path coefficients (βs) (see Figure 1) are shown in Table 4 and Table 5, as well as the standard cross-lagged model’s (Figure 2) results for observed variables only. The two models’ findings were very similar, and therefore we report the latent change score model’s only. Except for the paths from thrill- and adventure-seeking (baseline) to alcohol use (follow-up) (β_volume_ = −0.002, *p* = 0.861; β_binge_ = 0.011, *p* = 0.376) and from boredom susceptibility to volume of drinking (β_volume_ = 0.025, *p* = 0.059), all the paths from sensation seeking and its subscales to alcohol use were significant and positive. The path from experience seeking to binge drinking would become nonsignificant with Bonferroni-adjustment (*p* > 0.05). Paths from volume and binge drinking to thrill and adventure seeking would be at *p* = 0.05 level, whereby Mplus only presents three decimals and we cannot decide whether 0.005 is actually rounded down or up. Similarly, all paths from alcohol use to sensation seeking and its subscales were significant and positive, pointing to bidirectional associations between alcohol use and sensation seeking and its subscales. Interestingly, paths from alcohol use to sensation seeking and its subscales were higher than vice versa. Given the large sample size, these differences were significant at *p* < 0.001, except for thrill and adventure seeking (for binge: *p* =0.012; for volume: *p* = 0.0058). Hence, the latter would not be significant Bonferroni adjusted). Effect sizes reaching the convention for small effect sizes (EF > 0.1) were only found for paths from alcohol use variables to sensation-seeking (sub-)scales, but with the exception of disinhibition (and the total sensation-seeking scale for binge drinking) not for the paths from sensation seeking (sub-)scales to alcohol use variables. The largest, but still small, effect size was found for disinhibition (i.e., EF_binge-disinhibition_ = 0.181, and EF_disinhibition-binge_ = 0.163). 

## 4. Discussion

Personality traits are well known to have an impact on alcohol use, particularly externalizing personality traits, and among those, impulsivity may be the most relevant to alcohol use [3,10,11]. Among impulsigenic traits, meta-analyses have found sensation seeking to have the strongest association [3]. However, sensation seeking is not a single, homogeneous construct, but consists of different components. The present study indicated that the association between sensation seeking and alcohol use was mostly driven by disinhibition. In contrast, alcohol’s links to experience seeking and boredom susceptibility were much lower and partly even non-significant for thrill- and adventure-seeking. These findings confirmed an earlier review by Hittner and Swickert [17]. They also confirmed the findings from a small-scale, college student study by Lac and Donaldson [18], who found bidirectional effects for disinhibition and no significant effects for experience seeking and boredom susceptibility. 

Given sensation seeking’s prominent role in alcohol use and misuse, it is surprising that few studies have separated its different constructs to investigate their roles in alcohol use. However, doing so may result in different findings. For example, a meta-analytical review of the separate constructs of impulsivity and alcohol use [15] found that lack of perseverance, not sensation seeking, had the strongest impact on drinking quantity. Interestingly, studies analyzing disinhibition subscales were used for estimating the meta-analytical effect size of lack of perseverance, whereas studies using the thrill- and adventure-seeking subscales were used for estimating the sensation-seeking effect size. Under the UPPS model, sensation seeking does not comprise disinhibition or boredom susceptibility, which are aspects of lack of perseverance [3,12,15]. The study of Riley, Davis, Milich and Smith [19] actually dropped “sensation seeking”, which was measured as seeking out for thrilling stimulation, because no meaningful change over time could be found. This also confirm our findings of low or even non-significant associations with thrill and adventure seeking. The present study confirms, that it is important to look at subsconstructs to avoid “jingle” and “jangle” fallacies and to understand why some studies may find effects of sensations seeking whereas other does not. One aspect may be whether disinhibition is included or excluded in the concept of sensation seeking or whether sensation seeking basically represents thrill and adventure seeking. 

The reason why disinhibition is more strongly associated with alcohol use than the other constructs remains unclear. Sensation seeking is negatively associated with monoamine oxidase (MAO, [16]), thus heavy drinkers usually have lower MAO levels than lighter drinkers. MAO regulates levels of monoamines, such as dopamine, as well as dopaminergic reinforcement. Sensation seeking, thus, has been found to be positively associated with dopamine levels. As Hittner and Swickert [17] argued, one hypothesis why disinhibition is the sensation-seeking construct most strongly correlated with alcohol use could be that it has the strongest negative correlation with MAO. 

Although the link between sensation seeking and alcohol use is established, few studies have looked at longitudinal associations. For example, one meta-analysis on sensation seeking [17] indicated that 93% of the studies reviewed were cross-sectional. The MAO hypothesis would also help to explain the bidirectional association. Elevated dopamine levels may motivate alcohol use as a reward-seeking behavior, and alcohol use stimulates dopamine release (see [17], for this argument). Few studies have investigated the bidirectional association, partly because personality is seen as relatively stable over time [20], developing before alcohol use is initiated, and therefore rare longitudinal studies have focused on the prediction of alcohol use [27]. 

Our findings showed the importance of looking in both directions, confirming findings from college student or adolescent samples (Kaiser, Davis, Milich, Smith, and Charnigo [7]; White et al. [37]; Quinn, Stappenbeck, and Fromme [38]; Littlefield et al. [31]). Our findings particularly suggested the need to broaden thinking about the longitudinal paths from personality to alcohol so that it also includes paths from alcohol to personality change through sensation seeking subconstructs, as these pathways were generally larger in the present study. The bidirectional links found also agreed with the corresponsive principle [56], which states that two dynamics seen in individuals’ personalities and behaviors may support each other. Individuals may select environments (social selection) corresponding to their personality, and social influences, e.g., environmental experiences, affect personality. This can create a vicious circle. The study also supported new approaches in research on personality change, such as the TESSERA framework (Triggering situations, Expectancy, States/State expressions, and Reactions: [22]), which suggests that repeated short-term sequences of change, including triggering situations, expectancy, and associative processes (e.g., implicit learning, reinforcement learning, habit forming), may, through reinforcement and reward, lead to long-term changes in personality. Alcohol use seems to be an ideal candidate for such short-term sequences. 

## 5. Limitations

One of the present study’s major limitations is that it included only men. A meta-analytical review on traits related to impulsivity [3] and alcohol use found no gender differences in the associations, either for alcohol use per se or for problematic alcohol use. A meta-analysis on sensation-seeking subscales and alcohol use [17] found generally larger effect sizes for men than for women, particularly for experience-seeking, boredom susceptibility and thrill- and adventure-seeking, but not for disinhibition. This means that although future research should also include women, the present study’s findings for men alone may be indicative for women too. 

Although the present study’s effect sizes were small in conventional terms [54], studies of associations between personality change and other health-related changes, such as depression [57] or the onset of a chronic disease [58], showed similar or even smaller effect sizes. Indeed, the same was true for most studies on personality change and associated changes in alcohol use [7,40,59]. As argued by Riley, Davis, Milich and Smith [19], given the overall stability of personality over long developmental periods, the magnitude of personality change during shorter developmental periods are likely to be small. Even smaller effect sizes than those found for disinhibition are not meaningless. Hakulinen and Jokela [39] estimated that even much smaller effect sizes than ours might postpone normative personality changes by as much as three years over a five-year study period. One might expect changes (and effect sizes) during adolescence to be larger than those seen during early adulthood in the present study, as personality changes are smaller and alcohol use has become more normative (and habitual) among young adults than it is among adolescents. Another limitation is that only two measurement points could be used. More than two measurement points are needed to substantiate a vicious circle. The bidirectionality of our findings can only be indicative for the potential to create a vicious circle. As a final point, sensation seeking and subconstructs of it have also associations with other substances such as tobacco or illicit drugs, and therefore some shared variance with alcohol use, which could reduce effects with alcohol use. The inclusion of bidirectional associations with other substances would have rendered the paper and analysis highly complex and would not have been in the scope of the present paper. Moreover, meta-analyses of impulsivity and alcohol use commonly exclude studies who have measured other substances in addition to alcohol use [3,15] or did not include other substances as potential moderators [17]. The inclusion of other substances would therefore reduce comparability of our findings with these meta-analyses. It is believed, however, that similar analysis with the cross-domain coupling model may reveal new insights also for other substances. 

## 6. Conclusions

The present study described changes in alcohol use related to changes in sensation seeking, particularly disinhibition. Changes in personality traits related to sensation seeking may continue beyond adolescence and occur during emerging adulthood [21]. Given the bidirectional associations between personality and alcohol use, preventive efforts targeting both simultaneously might prove promising. Indeed, interventions like the *PreVenture* program, which tailor alcohol use interventions according to personality traits such as sensation seeking (while distinguishing it from impulsivity), anxiety, or negative thinking, have shown long-term effectiveness in several countries [32,33,34].

## Figures and Tables

**Figure 1 ijerph-19-12475-f001:**
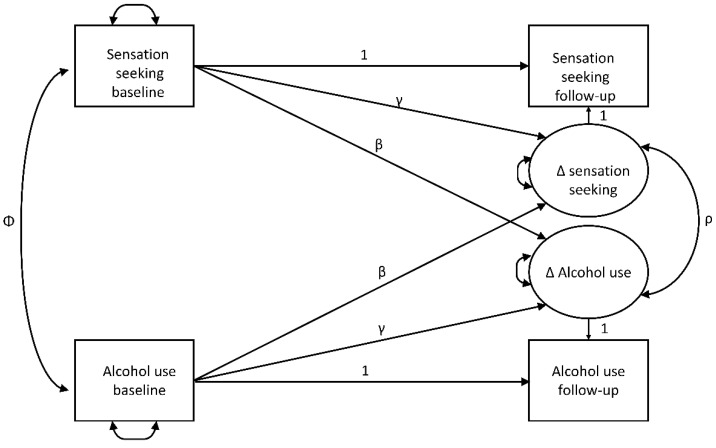
Theoretical Latent Change Score Model for Alcohol Use and Sensation-Seeking Traits. *Note.* Latent variables are drawn as circles. Manifest or measured variables are shown as squares. Residuals and variances are drawn as double-headed arrows into an object. Correlations are drawn as double-headed arrows between two objects (Φ, *ρ*). Paths (β, γ) are drawn as single-headed arrows. ∆s represent the latent change scores. 1s are unstandardized paths are fixed to 1 for identifyability of the model. Means are omitted for visual clarity. Models were calculated for two alcohol variables (binge and volume) and sensation seeking as total scale and for the four subscales.

**Figure 2 ijerph-19-12475-f002:**
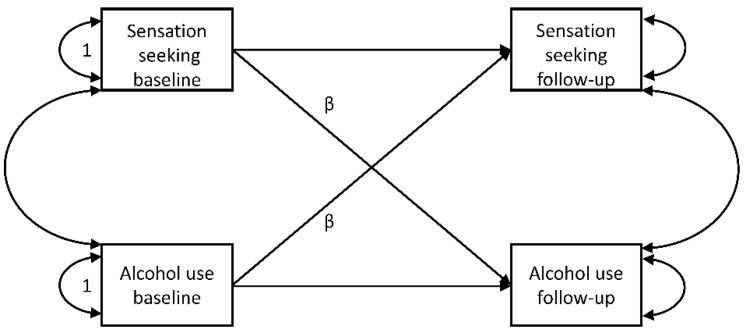
Theoretical Cross-lagged Model for Observed Variables of Alcohol Use and Sensation-Seeking Traits. *Note*. βs are the cross-lagged paths. 1s are unstandardized fixed variances. Models were calculated for two alcohol variables (binge and volume) and sensation seeking as total scale and for the four subscales.

**Table 1 ijerph-19-12475-t001:** Correlation matrix of all variables at both waves.

	SSSfull_BL	Experience_BL	Boredom_BL	Thrill_BL	Disinhibition_BL	Volume_BL	Binge_BL	SSSfull_FU	Experience_FU	Boredom_FU	Thrill_FU	Disinhibition_FU	Volume_FU
SSSfull_BL													
experience_BL	0.783												
boredom_BL	0.783	0.551											
thrill_BL	0.772	0.429	0.432										
disinhibition_BL	0.784	0.450	0.505	0.511									
volume_BL	0.315	0.159	0.223	0.157	0.461								
binge _BL	0.319	0.157	0.207	0.182	0.462	0.809							
SSSfull_FU	0.524	0.380	0.362	0.451	0.435	0.242	0.247						
experience_FU	0.391	0.433	0.270	0.244	0.269	0.133	0.139	0.785					
boredom_FU	0.347	0.268	0.362	0.210	0.253	0.149	0.141	0.752	0.540				
thrill_FU	0.445	0.224	0.224	0.606	0.307	0.126	0.142	0.758	0.412	0.378			
disinhibition_FU	0.416	0.240	0.270	0.290	0.507	0.340	0.340	0.775	0.449	0.453	0.484		
volume_FU	0.213	0.125	0.148	0.087	0.314	0.563	0.494	0.301	0.169	0.186	0.128	0.451	
binge _FU	0.218	0.113	0.145	0.104	0.330	0.527	0.513	0.302	0.162	0.177	0.136	0.460	0.791

*Notes*: BL: baseline; FU: Follow up; experience: SSSfull: total sensation seeking scale; experience: experience seeking; boredom: boredom susceptibility; thrill: thrill and adventure seeking; volume: logged volume in drinks per week; binge: binge drinking frequency. N varies between 5077 for the correlation between volume of drinking and disinhibition at baseline and 5121 for the correlation between volume of drinking and the total sensation seeking scale at follow-up.

**Table 2 ijerph-19-12475-t002:** Descriptive characteristics of changes in alcohol use and sensation seeking from baseline to follow-up.

		n	Baseline	Follow-Up	Difference Fu-BL	T-Value Difference	*p*-Value of *t*-Test	Correlation Baseline-Change	*p*-Value Correlation
Alcohol use	Drinks per week (log volume)	5099	1.68	1.59	−0.09	−6.66	<0.001	−0.509	<0.001
	Binge drinking frequency	5099	1.48	1.33	−0.15	−10.74	<0.001	−0.548	<0.001
Sensation-seeking	Total	5114	24.40	23.86	−0.53	−5.84	<0.001	−0.536	<0.001
	Experience-seeking	5107	6.82	7.09	0.27	8.09	<0.001	−0.565	<0.001
	Boredom susceptibility	5106	5.79	5.87	0.08	2.62	0.009	−0.611	<0.001
	Thrill- and adventure-seeking	5099	5.90	5.59	−0.31	−10.70	<0.001	−0.473	<0.001
	Disinhibition	5098	5.88	5.30	−0.58	−19.78	<0.001	−0.520	<0.001

*Note*: Fu-BL is follow-up measures minus baseline measures.

**Table 3 ijerph-19-12475-t003:** Correlations between observed values, residuals observed, and latent change scores for alcohol use and sensation seeking.

		n	Observed (Unadjusted)		Residuals Observed (Regression to the Mean)		Correlations between Latent Change Scores *	
			corr	*p*	corr	*p*	corr	*p*
Volume	Sensation-seeking total	5088	0.177	<0.001	0.207	<0.001	0.210	<0.001
	Experience seeking	5081	0.071	<0.001	0.107	<0.001	0.107	<0.001
	Boredom susceptibility	5080	0.107	<0.001	0.121	<0.001	0.123	<0.001
	Thrill- and adventure-seeking	5073	0.086	<0.001	0.086	<0.001	0.087	<0.001
	Disinhibition	5072	0.276	<0.001	0.324	<0.001	0.334	<0.001
Binge drinking	Sensation-seeking total	5088	0.163	<0.001	0.201	<0.001	0.204	<0.001
	Experience seeking	5081	0.065	<0.001	0.101	<0.001	0.101	<0.001
	Boredom susceptibility	5080	0.090	<0.001	0.113	<0.001	0.115	<0.001
	Thrill- and adventure-seeking	5074	0.085	<0.001	0.084	<0.001	0.083	<0.001
	Disinhibition	5072	0.257	<0.001	0.324	<0.001	0.336	<0.001

*Note:* * The latent change score model used Full Information Maximum Likelihood (FIML) estimation, taking participants with missing values into account under the ‘missing at random’ assumption; corr: correlation coefficient.

**Table 4 ijerph-19-12475-t004:** Standardized path coefficients of cross-lagged models of drinking volume with only observed (right) and observed and latent change score variables (left).

			Latent Change Score					Observed Variables Cross-Lagged Panel			
	BL → FU	Standard. Estimate	Lower 95% CI	Upper 95% CI	*p*-Value	Effect Size *	Standard. Estimate	Lower 95% CI	Upper 95% CI	*p*-Value	Approximate Effect Size *
Volume	Sensation-seeking total → volume	0.040	0.014	0.067	0.003	0.089	0.039	0.014	0.064	0.003	0.088
	Volume → sensation-seeking total	0.086	0.060	0.111	<0.001	0.134	0.087	0.061	0.112	<0.001	0.135
	Experience-seeking → volume	0.037	0.012	0.061	0.003	0.086	0.035	0.012	0.059	0.003	0.084
	Volume → experience-seeking	0.061	0.037	0.085	<0.001	0.110	0.066	0.040	0.092	<0.001	0.115
	Boredom susceptibility → volume	0.025	−0.001	0.046	0.059	0.075	0.024	−0.001	0.048	0.060	0.074
	Volume → boredom susceptibility	0.062	0.039	0.085	<0.001	0.111	0.073	0.046	0.100	<0.001	0.122
	Thrill- and adventure-seeking→ volume	−0.002	−0.027	0.023	0.861	−0.002	−0.002	−0.026	0.022	0.861	−0.002
	Volume → thrill- and adventure-seeking	0.036	0.011	0.062	0.005	0.085	0.033	0.010	0.056	0.005	0.082
	Disinhibition → volume	0.072	0.044	0.100	<0.001	0.121	0.069	0.042	0.960	<0.001	0.118
	Volume → disinhibition	0.136	0.108	0.163	<0.001	0.183	0.137	0.109	0.165	<0.001	0.184

*Notes*: * according to Peterson and Brown 2005 [55]. Both the cross-lagged panel model and the latent change score model used Full Information Maximum Likelihood (FIML), allowing for the inclusion of participants with missing values under the ‘missing at random’ assumption.

**Table 5 ijerph-19-12475-t005:** Standardized path coefficients of cross-lagged models of binge drinking with only observed (right) and observed and latent change score variables (left), continued.

			Latent Change Score					Observed Variables Cross-Lagged Panel			
	BL → FU	Standard. Estimate	Lower 95% CI	Upper 95% CI	*p*-Value	Effect Size *	Standard. Estimate	Lower 95% CI	Upper 95% CI	*p*-Value	Approximate Effect Size *
Binge	Sensation-seeking total → binge	0.059	0.034	0.084	<0.001	0.108	0.061	0.035	0.087	<0.001	0.110
	Binge → sensation-seeking total	0.089	0.064	0.114	<0.001	0.137	0.090	0.064	0.115	<0.001	0.138
	Experience-seeking → binge	0.032	0.008	0.056	0.009	0.081	0.033	0.008	0.057	0.009	0.082
	Binge → experience-seeking	0.067	0.043	0.090	<0.001	0.116	0.073	0.047	0.098	<0.001	0.122
	Boredom susceptibility → binge	0.040	0.015	0.064	0.001	0.089	0.041	0.016	0.066	0.001	0.090
	Binge → boredom susceptibility	0.059	0.036	0.081	<0.001	0.108	0.069	0.049	0.101	<0.001	0.118
	Thrill- and adventure-seeking → binge	0.011	−0.013	0.035	0.376	0.061	0.011	−0.014	0.036	0.377	0.061
	Binge → thrill- and adventure-seeking	0.036	0.011	0.061	0.005	0.085	0.032	0.010	0.055	0.005	0.081
	Disinhibition → binge	0.115	0.088	0.142	<0.001	0.163	0.118	0.091	0.145	<0.001	0.166
	Binge → disinhibition	0.134	0.107	0.161	<0.001	0.181	0.136	0.108	0.163	<0.001	0.183

*Notes*: * according to Peterson and Brown 2005 [55]. Both the cross-lagged panel model and the latent change score model used Full Information Maximum Likelihood (FIML), allowing for the inclusion of participants with missing values under the ‘missing at random’ assumption.

## Data Availability

All data of C-SURF are available via Zenodo under DOI: https://doi.org/10.5281/zenodo.5469953 (accessed on 29 September 2022).

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
