# Peer review of "Longitudinal Associations between Sensation Seeking and Its Components and Alcohol Use in Young SWISS Men—Are There Bidirectional Associations?"

_ijerph, 2022, doi:10.3390/ijerph191912475_

Round 1
Reviewer 1 Report
This is an interesting paper on an important topic. Strengths are the focus on bi-directionality, the large sample size, and the longitudinal design. The sample is well known and well described, cfr earlier publications.
If have only (very) minor comments:
- given the large number of comparisons I wonder whether correction for multiple comparisons is needed in the statistical analyses.
- The lack of including other substances of abuse might bias the results towards a specific role for alcohol. Indeed, e.g. smoking has been associated with both alcohol use and sensation seeking. leaving smoking (and no doubt a sizeable proportion of the participants smoke) limits in my view the interpretation of the data.
Reviewer 2 Report
The purpose of the paper was to investigate the association of sensation seeking with alcohol use and binge drinking in a large sample of young males who were followed for 5 years. Results point to bidirectional longitudinal associations between these variables. This study has several strengths, including the large sample recruited from the general population and the 5-year follow-up. However, I do have a number of concerns regarding the current manuscript, as described below:
1. One obvious limitation that is not mentioned in the limitations section is the availability of only 2 waves of measurement. In my perspective, this is not consistent with the notion of vicious circle included in the title, which would require at least 3 waves to determine if the mutual influences lead to substance use escalation. In addition, this leads to saturated SEM models, which precludes the evaluation of model fit.
2. More details should be provided regarding similarities and differences between the current study and the previous publication using the same cohort and the overall scores from the Brief Sensation Seeking Scale. If I am correct, the same waves and statistical models were used in that paper, so the value added from the current manuscript (based on the analysis of the BSSS subscales) is not entirely clear.
3. Please clarify how many conscripts were approached but refused participation.
4. Please clarify if only one AUDIT question was included in the measures. If the complete AUDIT was administered, please explain why it was not used in the current analyses.
5. The expression “highly significant” should be avoided, given that it can be confused by some readers with a large effect size.
6. Please replace Table 1 with a complete correlation table. This is part of the current minimum reporting standards for SEM in the field of psychology (Hoyle, R. H., & Isherwood, J. C. (2013). Reporting results from structural equation modeling analyses in Archives of Scientific Psychology. Archives of scientific psychology, 1(1), 14-22; Appelbaum, M., Cooper, H., Kline, R. B., Mayo-Wilson, E., Nezu, A. M., & Rao, S. M. (2018). Journal article reporting standards for quantitative research in psychology: The APA Publications and Communications Board task force report. American Psychologist, 73(1), 3-25).
7. In page 9, it is stated that “paths from alcohol use to sensation seeking and its subscales were higher than vice versa”. This statement could be formally tested by fixing both paths to be equal and determining if this restriction significantly reduces model fit.
8. In Table 4, only alcohol volume is reported, but binge drinking is not.
9. This paper seems relevant but is not referenced in the manuscript: Riley, E. N., Davis, H. A., Milich, R., & Smith, G. T. (2018). Heavy, problematic college drinking predicts increases in impulsivity. Journal of Studies on Alcohol and Drugs, 79(5), 790-798.
Author Response
Referee 2
We wish to thank the referee for the comments. We have addressed them as follows:
The purpose of the paper was to investigate the association of sensation seeking with alcohol use and binge drinking in a large sample of young males who were followed for 5 years. Results point to bidirectional longitudinal associations between these variables. This study has several strengths, including the large sample recruited from the general population and the 5-year follow-up. However, I do have a number of concerns regarding the current manuscript, as described below:
- One obvious limitation that is not mentioned in the limitations section is the availability of only 2 waves of measurement. In my perspective, this is not consistent with the notion of vicious circle included in the title, which would require at least 3 waves to determine if the mutual influences lead to substance use escalation. In addition, this leads to saturated SEM models, which precludes the evaluation of model fit.
Reply: We agree with the referee as regards the saturation of our models. We do not want to estimate parsimonious models or particular SEM models and evaluate the fit, but we want to estimate the cross-domain coupling model as indicated in the literature. This model is in fact saturated as would be many simple regression models or the cross-lagged panel models.
The referee is very right in pointing us to the lack of being able to fit vicious circles. We have changed the title accordingly. It now reads in the revised version as follows:
“Longitudinal associations between sensation seeking and its components and alcohol use in young Swiss men—are there bidirectional associations?”
In addition, we mentioned the limitation of only two waves in the limitation section as follows:
“Another limitation is that only two measurement points could be used. More than two measurement points are needed to substantiate a vicious circle. The bidirectionality of our findings can only be indicative for the potential to create a vicious circle.”
- More details should be provided regarding similarities and differences between the current study and the previous publication using the same cohort and the overall scores from the Brief Sensation Seeking Scale. If I am correct, the same waves and statistical models were used in that paper, so the value added from the current manuscript (based on the analysis of the BSSS subscales) is not entirely clear.
Reply: Our first paper analysed different general/overarching personality traits, namely aggression–hostility, sociability, neuroticism–anxiety, and impulsigenic personality trait operationalized with sensation seeking scale conceptualized as by Zuckermann and colleagues. In this paper we only used the full scale.
In between, we figured out that sensation seeking as part of impulsigenic personality trait has very different meanings and conceptualizations (e.g. in the UPPS model), which may explain that sensation seeking is sometimes associated with alcohol (and other substance) use and sometimes not (e.g. when sensation seeking is basically measured as thrill and adventure seeking or experience seeking). We believe that one reason is that disinhibition is not part of sensation seeking in e.g. UPPS model (the “jingle and jangle” fallacies). We therefore found it important to analyze different subconstructs of “sensation seeking”. In fact we found that disinhibition (in the UPPS part of lack of perseverance) but not thrill and adventure seeking (in the UPPS part sensation seeking) was related to alcohol use, which may explain some of the discrepancies found in the literature.
We completely revised the introduction (also using your helpful suggestion of Riley and colleagues) and strengthened the discussion towards this aspect of subconstructs of sensation seeking having differential impacts on alcohol use. We believe that the two papers, although using the same cohort data and statistical model have very different aims.
- Please clarify how many conscripts were approached but refused participation.
Reply: We added the following sentence:
“During enrollment procedures, 15,066 Swiss men showed up at the Army recruitment centers and were eligible for study inclusion. However, officers of the army were in charge to inform conscripts about the study. This was forgotten for few recruitment days resulting in a loss of 1,829 potential participants, who could not be approached for study participation. The loss is likely to be random.”
- Please clarify if only one AUDIT question was included in the measures. If the complete AUDIT was administered, please explain why it was not used in the current analyses.
Reply: We just wanted to measure binge drinking and not the full AUDIT. We also used the hint to the AUDIT to explain why we used the 6+ measure for standard drinks instead of the US measure of 5+ (both equal about 60+ grams of ethanol).
- The expression “highly significant” should be avoided, given that it can be confused by some readers with a large effect size.
Reply: Highly was just meant to mean p < 0.001; however we deleted “highly” and just referred to being significant.
- Please replace Table 1 with a complete correlation table. This is part of the current minimum reporting standards for SEM in the field of psychology (Hoyle, R. H., & Isherwood, J. C. (2013). Reporting results from structural equation modeling analyses in Archives of Scientific Psychology. Archives of scientific psychology, 1(1), 14-22; Appelbaum, M., Cooper, H., Kline, R. B., Mayo-Wilson, E., Nezu, A. M., & Rao, S. M. (2018). Journal article reporting standards for quantitative research in psychology: The APA Publications and Communications Board task force report. American Psychologist, 73(1), 3-25).
Reply: We have replaced table 1 by the full correlation matrix.
- In page 9, it is stated that “paths from alcohol use to sensation seeking and its subscales were higher than vice versa”. This statement could be formally tested by fixing both paths to be equal and determining if this restriction significantly reduces model fit.
Reply: Thanks for the comment. We followed the advice and tested the differences between the two paths. All differences were significant. We changed the Method section as follows:
“βs can be interpreted as cross-lagged coefficients (coupling parameters). They indicate, whether subscales of sensation seeking and the full scale (baseline) predict alcohol use change scores (follow-up) or whether alcohol use (baseline) predicts personality change scores. The difference was tested by fixing the coefficients to be equal and comparing the model fit versus the unrestricted model.”
Reply: We also added the following to the results section:
“Interestingly, paths from alcohol use to sensation seeking and its subscales were higher than vice versa. Given the large sample size, these differences were significant at p < 0.001, except for thrill and adventure seeking (for binge: p =0.012; for volume: p = 0.0058). Hence, the latter would not be significant Bonferroni adjusted).”
- In Table 4, only alcohol volume is reported, but binge drinking is not.
Reply: We want to thank the referee for pointing us to this flaw. In fact, we usually have all tables at the end of a manuscript Table 4 went over two pages. The style of IJERPH requests to have tables integrated in the text body. Mistakenly we copied only the first half of the table. This has now been corrected in the revised version of the manuscript.
- This paper seems relevant but is not referenced in the manuscript: Riley, E. N., Davis, H. A., Milich, R., & Smith, G. T. (2018). Heavy, problematic college drinking predicts increases in impulsivity. Journal of Studies on Alcohol and Drugs, 79(5), 790-798.
Reply: We are grateful for this reference. It perfectly fits in our argumentation. We used it at three different places:
- In the Introduction as further example of the “jingle and jangle fallacies”:
“For example, a longitudinal study of Riley, et al. [19] using the UPPS conceptualization could not find any meaningful change in sensation seeking among college freshman, and therefore the authors did not model its association with alcohol use. Sensation seeking was defined by thrill and adventure seeking. However, the authors found that alcohol problems predicted changes in urgency, lack of planning, and lack of perseverance, which contains aspects of sensation seeking (e.g. disinhibition or boredom susceptibility) as defined by Zuckermann et al. [13].” - In the Discussion:
“The study of Riley, Davis, Milich and Smith [19] actually dropped “sensation seeking”, which was measured as seeking out for thrilling stimulation, because no meaningful change over time could be found. This also confirm our findings of low or even non-significant associations with thrill and adventure seeking.” - In the Limitation section justifying small effect sizes:
“As argued by Riley, Davis, Milich and Smith [19], given the overall stability of personality over long developmental periods, the magnitude of personality change during shorter developmental periods are likely to be small.”
